# Usefulness of Endoscopy for the Detection and Diagnosis of Primary Esophageal Motility Disorders and Diseases Relating to Abnormal Esophageal Motility

**DOI:** 10.3390/diagnostics13040695

**Published:** 2023-02-12

**Authors:** Shiko Kuribayashi, Hiroko Hosaka, Toshio Uraoka

**Affiliations:** Department of Gastroenterology and Hepatology, Gunma University Graduate School of Medicine, Maebashi 371-0851, Japan

**Keywords:** endoscopic findings, esophageal motility disorders, image-enhanced endoscopy

## Abstract

Esophagogastroduodenoscopy (EGD) is performed to rule out organic diseases in the diagnosis of esophageal motility disorders (EMDs). Abnormal endoscopic findings can be observed during EGD, which indicate the presence of EMDs. Several endoscopic findings at both the esophagogastric junction and esophageal body that are related to EMDs have been reported. Gastroesophageal reflux disease (GERD) and eosinophilic esophagitis (EoE) could be detected during EGD, and these diseases are often associated with abnormal esophageal motility. Image-enhanced endoscopy (IEE) could improve the detection of these diseases during EGD. Although no report has been published previously on the potential usefulness of IEE in the endoscopic diagnosis of EMDs, IEE can be used to detect disorders that can be associated with abnormal esophageal motility.

## 1. Introduction

Esophageal motility disorders (EMDs) can cause dysphagia or chest pain and are diagnosed by esophageal manometry or esophagram. Although esophagogastroduodenoscopy (EGD) is performed in the diagnosis of EMDs to rule out organic diseases, such as esophageal cancer or gastroesophageal reflux disease (GERD), typical endoscopic findings are sometimes seen during EGD in patients with EMDs that could indicate the presence of EMDs [1].

EMDs are classified as either primary or secondary EMDs. The Chicago classification classified primary EMDs based on characteristics of esophageal motility abnormalities as achalasia, esophagogastric junction outflow obstruction, distal esophageal spasm, hypercontractile esophagus, absent contractility, or ineffective esophageal motility [2]. Distal esophageal spasm and hypercontractile esophagus are known as spastic EMDs.

The utility of image-enhanced endoscopy (IEE) in both the detection and evaluation of esophageal cancers, including Barrett’s adenocarcinoma [3,4,5,6,7], has been well established. Narrow band imaging (NBI), flexible spectral imaging color enhancement (FICE) [8,9], blue laser imaging (BLI) [10], linked color imaging (LCI) [11] and i-scan [12] are currently used. Several papers discuss the usefulness of IEE in the detection of non-malignant diseases, such as GERD.

Nowadays, per-oral endoscopic myotomy (POEM) is widely performed in patients with achalasia [13]. Not only short-term but also long-term outcomes of POEM have been published [14,15,16]. In addition, the usefulness of POEM in patients with non-achalasia spastic motility disorders has been reported, and the European Society of Gastrointestinal Endoscopy (ESGE) recommends POEM for spastic motility disorders other than achalasia [17].

Thus, endoscopic treatment is useful to manage achalasia or spastic esophageal motility disorders. However, it is not known whether endoscopy is useful in the diagnosis of EMDs and whether IEE is useful in diagnosing EMDs. In this review, we focused on the usefulness of endoscopy for the detection and diagnosis of EMDs.

## 2. Detection or Diagnosis

### 2.1. Endoscopic Findings Related to Primary EMDs

Several endoscopic findings are often seen in patients with achalasia: dilated esophagus, tortuous esophagus, presence of residue and functional stenosis of the esophago-gastric junction (EGJ), which means resistance when the scope is passing through at the EGJ [1,18,19]. The presence of “foam” in the esophagus suggests the presence of EMDs but is not a specific finding of EMDs [1]. Esophageal rosette represents impaired EGJ relaxation [20]. In healthy subjects the esophageal palisade vessels can be seen during a deep inspiration; however, the full extent of the esophageal palisade vessels cannot be observed even during the deep inspiration and rosette-like appearance is often seen at the EGJ in patients with achalasia (Figure 1). Recently, Gingko leaf sign has been reported as a new endoscopic finding of achalasia [21]. The full extent of the esophageal palisade vessels cannot be seen, and a Gingko leaf-shaped morphology of a longitudinal section of the EGJ at the end of a deep inspiration is sometimes seen in patients with achalasia in whom esophageal rosette cannot be seen (Figure 2). Champagne glass (CG) sign is an endoscopic finding that suggests the presence of impaired EGJ relaxation [22]. The CG sign is defined as the failure of the proximal border of the lower esophageal sphincter (LES) to open; however, the abdominal esophagus at the squamocolumnar junction (SCJ) is open in the retroflex view from the stomach (Figure 3 and Figure 4). CG can be classified into two types: CG-1 is when the distance from the SCJ to the proximal border of the LES is less than one centimeter, and CG-2 is when the same distance is more than one centimeter. The CG sign was seen in 71.3% of patients with achalasia; CG-1 in 65.1% and CG-2 in 6.1% [22].

A pinstripe pattern in the esophageal body has been reported as an endoscopic finding in patients with achalasia in the early stages [23]. The pinstripe pattern is defined as the presence of longitudinal superficial wrinkles of the esophageal mucosa (Figure 5). The pinstripe pattern was observed in 60.7% of achalasia patients overall, and it was observed in 62.5% of patients with a history shorter than 10 years.

A corona appearance has also been reported as an endoscopic finding indicating impaired EGJ relaxation [24]. The corona appearance is defined as positive when all the following three criteria are met upon observation of the LES performed with the attached short-type ST hood: (a) congestion inside the hood, (b) ischemic change around the hood, and (c) palisade vessels outside the hood. The corona appearance had the highest sensitivity compared with other endoscopic findings in the study, such as functional stenosis of the EGJ, mucosal thickening and whitish change, abnormal contraction of the esophageal body, dilation of the esophageal lumen and food remnant. Since a short-type ST Hood is necessary for evaluating the presence of the corona appearance, it cannot be observed during the routine EGD.

Regarding endoscopic findings in patients with non-achalasia spastic EMDs, corkscrew esophagus is known as an endoscopic finding that represents abnormal esophageal contraction (Figure 6) [25,26]. It can be seen in patients with diffuse esophageal spasm or hypercontractile esophagus [27]. In a multicenter cohort study in Japan with 87 patients with Jackhammer esophagus, abnormal endoscopic findings were detected in 32% of the patients [28]. Corkscrew or rosary beads appearance was seen in 26% of the patients and esophageal narrowing was present in 11% of the patients. Although the corkscrew appearance is a typical endoscopic finding that indicates the presence of EMDs, there is often no abnormality observed during EGD in patients with a hypercontractile esophagus.

Evaluation of weak contractions in the esophageal body is challenging. The esophageal lumen is occluded during esophageal peristalsis in subjects with normal esophageal motility. It has been reported that non-occlusive contraction could indicate weak peristalsis [29,30].

### 2.2. Endoscopic Findings That Could be Associated with EMDs

#### 2.2.1. Diverticulum or Esophageal Intramural Pseudodiverticulosis

Esophageal diverticulum could indirectly suggest the presence of EMDs [31,32,33]. Intraluminal pressure in the esophagus increases due to esophageal dysmotility, which could cause a diverticulum. An esophageal diverticulum due to pulsion could occur in any part of the esophagus. Epiphrenic diverticulum can be observed just above the EGJ [34].

Esophageal intramural pseudodiverticulosis (EIPD) is a rare condition in which multiple small outpouchings are seen in the wall of the esophagus. EIPD could be associated with esophageal motility disorders, such as achalasia [35].

#### 2.2.2. Candida Esophagitis

Candida esophagitis could be observed in patients with EMDs [36]. Stasis of fluid or food in the esophagus could cause candidiasis. Although there are many causes of candida esophagitis in addition to EMDs, candida esophagitis with stasis could be a sign of EMD. It should be noted that candida esophagitis could worsen the symptoms and prevent the detection of esophageal cancer in patients with EMDs.

### 2.3. Identification of Primary EMDs during EGD

It has been reported that only 38% of patients with achalasia who were diagnosed by esophageal manometry were diagnosed correctly by EGD [37]. However, careful endoscopic observation can raise the accuracy of endoscopic diagnosis of EMDs. When 20 patients with achalasia, 13 patients with systemic sclerosis and 33 controls were evaluated, the accuracy of endoscopic diagnosis of esophageal motility was 96% [29]. When 380 patients with dysphagia who underwent high-resolution esophageal manometry (HREM) were evaluated, the incidence of abnormal endoscopic findings was 64.4% [30]. In the study functional stenosis of the EGJ, residue in the esophagus and non-occlusive contraction were significantly associated with EMDs. When 273 patients who underwent HREM were evaluated, abnormal endoscopic findings were not observed during EGD in 41 out of 273 patients (15%) [38].

### 2.4. Endoscopic Findings of Diseases That Could Be Related to Abnormal Esophageal Motility

#### 2.4.1. GERD

GERD can be divided into three phenotypes based on endoscopic findings: reflux esophagitis (RE), non-erosive reflux disease (NERD) and Barrett’s esophagus (BE). Mucosal breaks are seen in endoscopy in RE, while they are not seen in NERD, although typical reflux symptoms are present. BE is characterized as columnar epithelium in the esophagus.

GERD is associated with abnormal esophageal motility [39]. Longer acid exposure time (AET) in the supine position was observed in patients with abnormal esophageal body motility [40,41]. Increasing severity of GERD is associated with decreasing resting EGJ pressure and amplitude of the distal esophagus, higher prevalence of hiatal hernia and increased prevalence of ineffective esophageal motility (IEM) [42]. Fragmented peristalsis is often seen in patients with GERD, especially in those with BE [43]. Since patients with BE have low resting EGJ pressure [44], low amplitude of peristalsis and frequent failed peristalsis [45], higher acid exposure in the esophagus was observed [46]. The severity of esophageal motility abnormalities in patients with BE could be comparable to those in patients with severe reflux esophagitis [47]. In addition, ineffective esophageal contraction in patients with BE could persist on provocative tests, such as rapid drink challenge or solid meal [48]. Primary peristalsis in patients with NERD could be comparable to that in subjects without GERD; however, the triggering of secondary peristalsis could be defective in patients with NERD, which could lead to prolonged AET [49]. It has been reported that the vigor of peristalsis during multiple rapid swallows was inversely correlated with AET in patients with NERD [50]. Moreover, esophageal motility abnormalities could be detected during a solid meal test although there was no abnormal esophageal motility during liquid swallows in patients with NERD [51].

The LA classification has been proposed to assess the severity of RE [52]. This classification has been validated and is widely used across the world [53]. Although there are observer variations that depend on the level of endoscopic experience [54], the reproducibility of the classification was good for both expert and trainee endoscopists in a study in which inter- and intra-observer consistency of the classification was assessed [55].

Minimal change (LA Grade M) is often used in Japan. LA grade M is defined as erythema without sharp demarcation, whitish turbidity and/or invisibility of vessels [56,57,58]. It has been reported that minimal changes were observed in patients with reflux symptoms more frequently than in those without reflux symptoms [59]. Moreover, acid reflux events detection by 24 h esophageal pH monitoring was significantly higher in patients with minimal change than in controls [60]. However, inter-observer agreement on the minimal change is poor among endoscopists [52,61].

IEE could improve the detection of GERD. Several endoscopic features regarding GERD have been reported: 1) Increased number, dilatation and tortuosity of intrapapillary capillary loops (IPCLs), 2) Presence of microerosions, 3) Vascularity at the SCJ, 4) Presence of columnar island in the distal esophagus, and 5) Ridge-villous pattern above the SCJ characterized by the presence of uniform, longitudinally aligned ridges alternating with a villiform pattern [62]. Among these endoscopic findings, changes in IPCLs and microerosions, and increased vascularity at the SCJ were significant for detecting GERD. Sensitivity and specificity in changes of IPCLs were 60–80%, and those in microerosions and increased vascularity at the SCJ were 40–50% and 90–100%, respectively. A recent randomized, controlled trial showed that ridge-villous patterns above the SCJ on NBI demonstrated high specificity, correlated with AET, and improved with proton pump inhibitors in patients with NERD [63]. In addition, NBI can provide inter-observer and intra-observer consistency in grading RE [64,65]. I-scan could improve the identification of minimal changes in patients with NERD [66,67]. Sensitivity (51.35%), specificity (67.33%), positive predictive value (PPV, 36.54%) and negative predictive value (NPV, 79.06%) of minimal changes detected by i-scan in detecting GERD confirmed by the presence of RE or abnormal AET in 24 h esophageal impedance-pH monitoring [68]. FICE could provide higher sensitivity, NPV and accuracy than white light imaging (WLI) [69]: Sensitivity, specificity, PPV, NPV and accuracy of FICE were 77.8%, 83.3%, 93.3%, 55.6% and 79.2%, respectively. However, the inter-observer agreement was poor. LCI can improve the detection of minimal changes in patients with NERD [70] and the visibility of RE [71]. However, inter-observer agreements in diagnosing RE, including minimal changes in BLI and LCI, were not high [72].

Since BE is a premalignant condition, it is important to accurately recognize BE. It is known that the length of BE is associated with a risk for esophageal adenocarcinoma. In addition, it is also known that the length of BE is correlated with AET [73]. The Prague C & M criteria have been developed to standardize the endoscopic grading of BE [74]. The “C” value represents the circumferential extent, and the “M” value represents the maximal extent. The overall reliability coefficients for the assessment of the C & M extent of the endoscopic BE segment above the EGJ were 0.95 and 0.94, respectively. The rates of exact agreement for C & M values for pairwise comparisons of individual patient values were 53% and 38%, respectively. The overall reliability coefficient for endoscopic recognition of BE equal to, or more than, 1 cm was 0.72, whereas it was 0.22 for BE less than 1 cm. The reliability coefficients for recognizing the location of the EGJ and the diaphragmatic hiatus were 0.88 and 0.85, respectively. Although this study was performed by experts, the criteria were also validated in trainees [75]. The overall intraclass correlation coefficients for assessment of the C & M extent of the endoscopic BE segment above the GEJ were 0.94 and 0.96, respectively. The overall intraclass correlation coefficients for GEJ and diaphragmatic hiatus location recognition were 0.92 and 0.90, respectively. Moreover, duration of training did not affect the inter-observer agreement.

It should be recognized that there are different definitions of BE among different guidelines [76,77,78,79,80,81]. Some guidelines require the minimal length of columnar epithelium from the EGJ to be at least 1 cm [77,79,80,81], whereas others do not have any limitation on the length of the columnar epithelium [76,78]. Some guidelines require the presence of intestinal metaplasia for diagnosing BE [76,79,81], but others do not [77,78,80]. The definition of EGJ is also different among guidelines. The proximal end of gastric folds defines the EGJ in some guidelines [76,77,79,80,81]; however, the distal end of palisade vessels defines the EGJ in other guidelines [78]. Recently, an international consensus report on the anatomy, pathophysiology and clinical significance of the EGJ has been published [82]. The consensus report recommends that the length of BE and the presence of intestinal metaplasia should not be required for diagnosing BE. The distal end of palisade vessels is preferred for use as an endoscopic landmark of the EGJ.

Although it is not an aim of this review, it is important to detect esophageal adenocarcinoma in patients with BE. Since surveillance endoscopy could contribute to detecting early-stage esophageal adenocarcinoma [83], surveillance endoscopy is recommended by the American Society for Gastrointestinal Endoscopy (ASGE) [84]. IEE is useful to detect esophageal adenocarcinoma in patients with BE. A systematic review and meta-analysis showed that target biopsies with chromoendoscopy, NBI, and endoscope-based confocal laser endomicroscopy met the thresholds of the ASGE preservation and incorporation of valuable endoscopic innovations (PIVI) on imaging technology when experts in advanced imaging techniques use these techniques [85]. Therefore, the use of chromoendoscopy or virtual chromoendoscopy, such as NBI, in addition to WLI has been recommended during the surveillance of BE [84].

#### 2.4.2. Eosinophilic Esophagitis

EMDs are often seen in patients with eosinophilic esophagitis (EoE) [28,86]. Interestingly, not only IEM but also hypercontractile esophagus can be observed. Esophageal motility is associated with disease severity in patients with EoE [87].

An assessment of eosinophil infiltration in the esophageal mucosa is necessary for diagnosing EoE [88,89]. Distinctive endoscopic findings, such as edema, rings, exudates, furrows and stricture are often observed in patients with EoE. It has been shown that the distribution of eosinophil infiltration in the esophageal mucosa is heterogenous, and more intense eosinophil infiltration could be observed at the sites of those distinctive endoscopic findings [90]. Therefore, biopsy from appropriate sites of the esophagus, as determined by careful endoscopic evaluation, is important for diagnosing EoE [91]. A meta-analysis revealed that there is substantial heterogeneity in the prevalence of EoE among studies [92]. In this meta-analysis, 17% of patients had endoscopically normal findings; however, the prevalence of normal endoscopic findings decreased when the analysis was limited to prospective studies. The overall sensitivity of determining endoscopic features was modest, ranging from 15% to 48%, whereas levels of specificity were greater, ranging from 90 to 95%. In addition, the accuracy of predicting positive values ranged from 51% to 73%, and that of predicting negative values ranged from 74% to 84%. Although the sensitivity of detecting endoscopic features was not sufficient, at least one abnormal endoscopic finding was detected during EGD in 93% of patients in the meta-analysis. Recently, the EREFS classification system has been proposed for grading these endoscopic findings [93]. The EREFS system identifies patients with EoE with an area under the receiver operator characteristic curve of 0.934 [94]. Furthermore, the score decreases with treatment, and histologic responders have significantly lower scores than nonresponders. A caterpillar sign is defined as a fragile, protruding mucosal lesion sandwiched between longitudinal furrows. The sensitivity, specificity, positive predictive value and negative predictive value of the caterpillar sign for the diagnosis of EoE were 83.3%, 98.1%, 96.2% and 91.2%, respectively [95]. Furthermore, inter-observer agreement in the identification of the caterpillar sign was substantial (κ = 0.80). Large esophageal laceration due to fragility of the esophageal mucosa could occur, which is called “crêpe-paper” mucosa [96]. The specificity of determining endoscopic features of EoE is sufficient; however, the sensitivity of determining those features is not yet sufficient. Therefore, the diagnostic utility of determining endoscopic features is still unsatisfactory, and many biopsies from multiple sites in the esophagus have been required even if there are no typical endoscopic features of EoE.

The usefulness of IEE in diagnosing EoE has been reported. NBI with magnification is useful to evaluate esophageal mucosa in patients with EoE. Three endoscopic findings were observed: beige mucosa, dot-shaped IPCL and absent cyan vessels [97]. In addition, dilation, shortening and congestion of IPCL, and granular surface without erosion were also reported as endoscopic findings in EoE on NBI with magnification [98]. Recently, it has been reported that beige mucosa on NBI without magnification was also useful to detect EoE (Figure 7 and Figure 8) [99]. The sensitivity, specificity and overall accuracy of beige mucosa in predicting EoE activity were 97.8%, 96.9% and 97.8%, respectively. LCI could be useful in the endoscopic diagnosis of EoE. For example, a case was reported in which typical endoscopic findings, such as furrows, rings, or exudates were not clearly observed; however, an edematous mucosa was detected as a yellowish area on LCI [100]. Furthermore, LCI could contribute to improving the detection of typical endoscopic findings [101]. Combined WLI and LCI had higher accuracy for diagnosing EoE endoscopically (0.85 vs. 0.70) than WLI alone. Inter-observer agreement on combined WLI and LCI was higher than that of WLI alone.

#### 2.4.3. Systemic Sclerosis or Other Autoimmune Diseases

Esophageal motility abnormalities are often seen in patients with systemic sclerosis [102]. IEM and absent contractility are common EMDs; however, various EMDs, such as esophagogastric junction outflow obstruction, distal esophageal spasm, or hypercontractile esophagus, could be seen [103]. Poor esophageal clearance due to esophageal motility abnormalities could lead to severe esophagitis [104] and could prolong AET even on high-dose proton pump inhibitors (PPIs) [105].

Cameron et al. evaluated the motility of both the esophageal body and EGJ during EGD. They found that EGJ did not close in 12 out of 13 patients with systemic sclerosis [29]. In addition, lumen-occluding peristaltic contractions were identified in the middle esophagus in only 3 out of 13 patients. In recording a diagnostic sequence of EGJ opening followed by esophageal body contraction and subsequent EGJ closing, only 1 out of 13 patients with systemic sclerosis had normal esophageal motility in the study. Based on these results, they concluded that esophageal motility in patients with systemic sclerosis could be evaluated by EGD, and that assessment of esophageal motility during EGD may represent an adjunctive diagnostic test to manometry.

Esophageal motility abnormalities could be observed in patients with not only systemic sclerosis but also other autoimmune diseases such as mixed connective tissue disease (MCTD) [106], Sjögren’s syndrome [107,108,109], systemic lupus erythematous (SLE) [110], rheumatoid arthritis (RA) [111] and idiopathic inflammatory myopathy [112]. Involvement of the esophagus is seen in up to 85% of patients with MCTD [113]. The pattern of esophageal motility abnormalities in patients with MCTD is similar to that in systemic sclerosis; however, dysfunction of the upper esophageal sphincter is often seen [106]. Decreased or absent contractility in the upper third of the esophagus is observed in patients with Sjögren’s syndrome [114], and the upper esophageal sphincter impairment in these patients may be more severe than that in other connective tissue diseases [115]. Decreased esophageal motility could be observed in up to 72% of patients with SLE [116]. Although esophageal motility abnormalities are mostly mild, absent contractility may be seen in 10–25% of patients with SLE [110,117]. Similar to the findings in Sjögren’s syndrome, esophageal motility in the upper third of the esophagus is often impaired in these patients [118]. Esophageal motility abnormalities could be seen in approximately 30% of patients with RA [119]. It has been reported that esophageal motility abnormalities are common in patients with idiopathic inflammatory myopathy, and both the proximal and distal esophagus could be involved [120]. In addition, the degree of distal esophageal involvement seems to correlate with the duration of myositis.

#### 2.4.4. Systemic Amyloidosis

Various organs are involved in patients with systemic amyloidosis, and the esophagus could also be involved. Esophageal motility abnormalities can be seen in patients with systemic amyloidosis [121]. Esophageal dysmotility could be caused by myopathic and/or neuropathic involvements. A study, in which a total of 30 patients with systemic amyloidosis (24 patients with primary amyloidosis and 6 patients with secondary amyloidosis) were studied, showed that resting EGJ pressure was decreased in 12 patients with primary amyloidosis (50%) and 2 patients with secondary amyloidosis (33%). In addition, motility abnormalities in the esophageal body were seen in nine patients with primary amyloidosis (38%) and a patient with secondary amyloidosis (17%). There were no patients with abnormalities of the upper esophageal sphincter. Various esophageal motility abnormalities could be seen. A case of a patient with systemic amyloidosis who had hypertensive EGJ and impaired EGJ relaxation has been reported [122,123,124].

Amyloid deposition in the gastrointestinal tract is different between types of amyloidosis. Amyloid light-chain (AL) deposition is seen in the submucosal layer and/or muscularis propria in patients with AL amyloidosis, whereas serum amyloid A protein (AA) deposition is seen on the lamina propria mucosae and submucosal layer in patients with AA amyloidosis [125]. Therefore, endoscopic findings are more likely to be observed in patients with AA amyloidosis than those with AL amyloidosis. Endoscopic findings of systemic amyloidosis are not specific, and various endoscopic findings, such as fine granular appearance, mucosal friability, erosions, ulcerations, erythema, or polypoid protrusions could be observed. A report showed that 31 of 37 patients with systemic amyloidosis (8 AL amyloidosis and 29 AA amyloidosis patients) had normal endoscopic findings on EGD.

#### 2.4.5. Von Recklinghausen’s Neurofibromatosis

Neurofibroma is a benign neoplasia that consists of neural and connective tissues. The prevalence of neurofibroma is estimated to represent approximately 0.9% of esophageal submucosal tumors [126]. Visceral organ neurofibromas are associated with genetic disorders such as Von Recklinghausen’s disease [127]. Although gastrointestinal involvement can be found in the small bowel or colon, the esophagus could be involved in patients with Von Recklinghausen’s disease. It is known that Von Recklinghausen’s neurofibromatosis could cause EMDs, such as achalasia. Hypoganglionosis was seen in a patient with Von Recklinghausen’s disease who presented achalasia [128]. It is hypothesized that hypoganglionosis may have resulted from transsynaptic degeneration or from abnormal embryological development of neural crest cells concluding that disturbance of gastrointestinal motility may occur in neurofibromatosis, but the underlying mechanism is not clear. A case has also been reported in which there were multiple neurofibromas at the EGJ, which could cause achalasia [129]. Thus, if symptoms related to EMDs, such as dysphagia, are reported in patients with Von Recklinghausen’s neurofibromatosis, EGD should be performed to detect EMDs as well as esophageal submucosal tumors.

## 3. Conclusions

Abnormal endoscopic findings could be observed during EGD in patients with EMDs. Although no report has been published that assesses the potential usefulness of IEE in the endoscopic diagnosis of EMDs, IEE can be used to detect disorders that can be associated with abnormal esophageal motility.

## Figures and Tables

**Figure 1 diagnostics-13-00695-f001:**
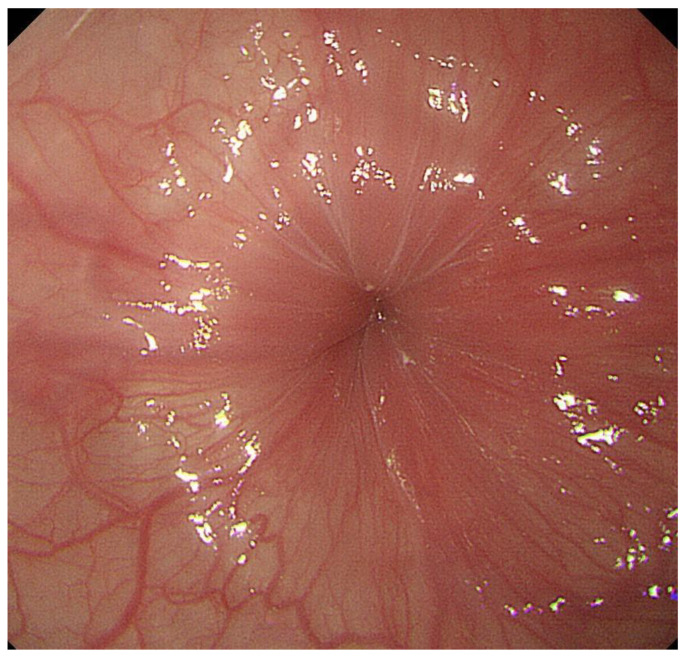
A case in which esophageal rosette was observed. Rosette-like appearance was observed at the esophago-gastric junction during deep inspiration.

**Figure 2 diagnostics-13-00695-f002:**
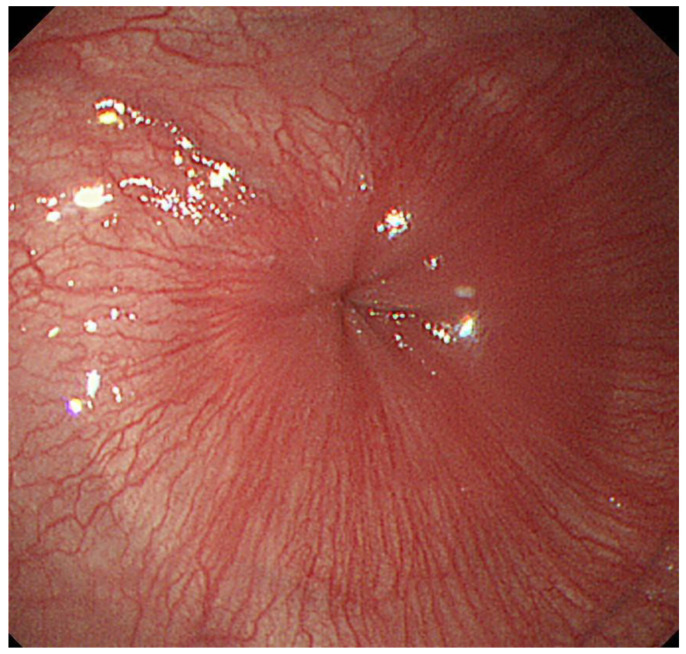
A case in which Gingko leaf sign was observed. Gingko leaf-shaped morphology was observed at the esophago-gastric junction during the end of deep inspiration.

**Figure 3 diagnostics-13-00695-f003:**
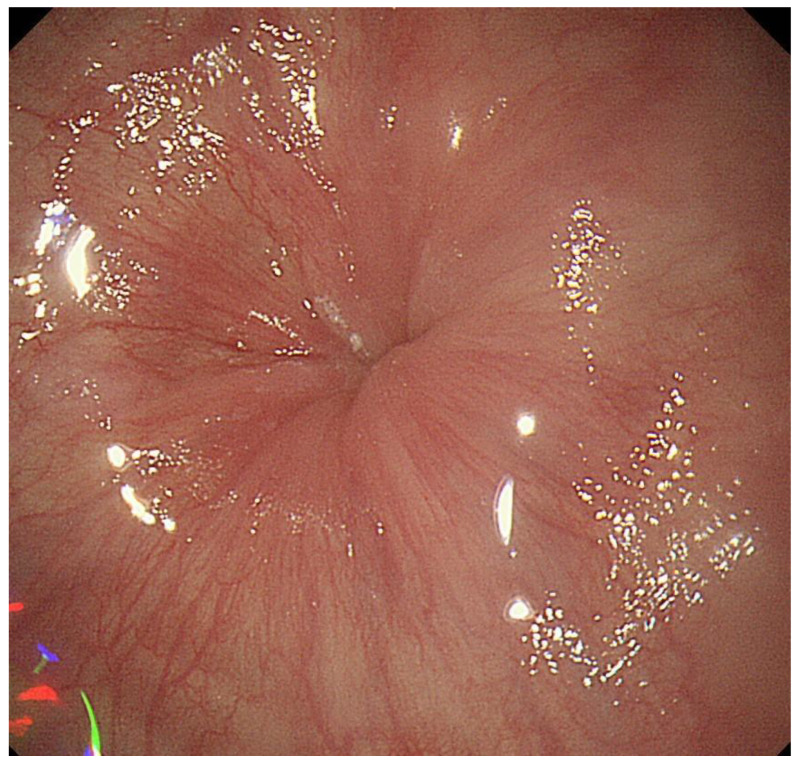
A case with achalasia in which Champagne glass sign was observed. Gingko leaf sign was observed at the esophago-gastric junction during deep inspiration.

**Figure 4 diagnostics-13-00695-f004:**
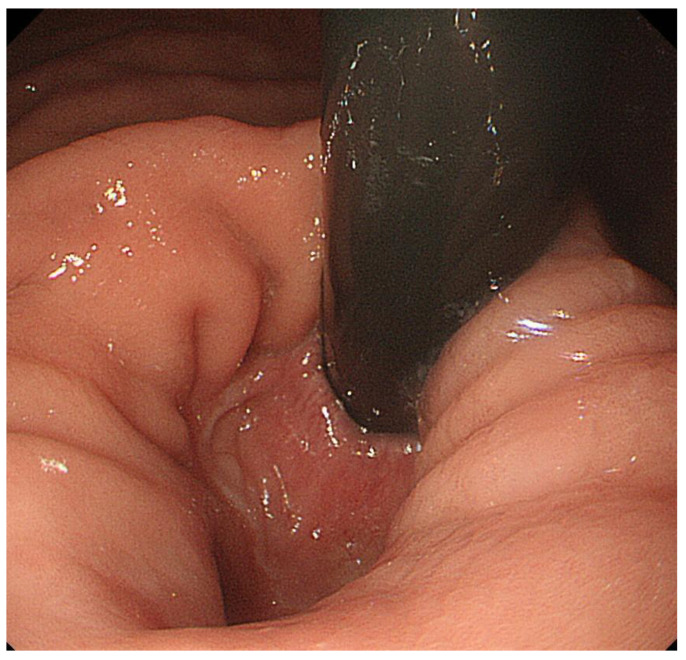
Although the squamocolumnar junction could not be observed even during the deep inspiration in the anterograde view from the esophagus (Figure 3), it could be observed in the retroflex view from the stomach.

**Figure 5 diagnostics-13-00695-f005:**
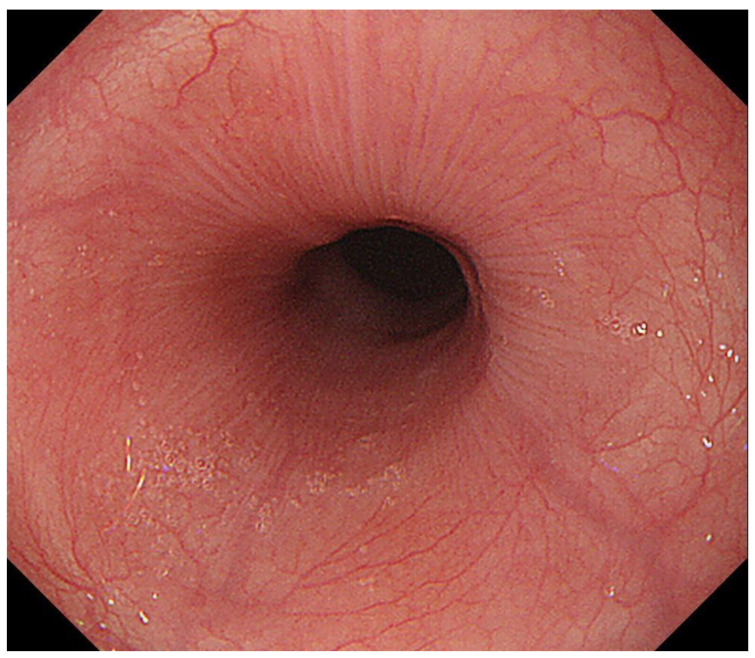
A case with achalasia in which pinstripe pattern was observed. Longitudinal superficial wrinkles of the esophageal mucosa were observed.

**Figure 6 diagnostics-13-00695-f006:**
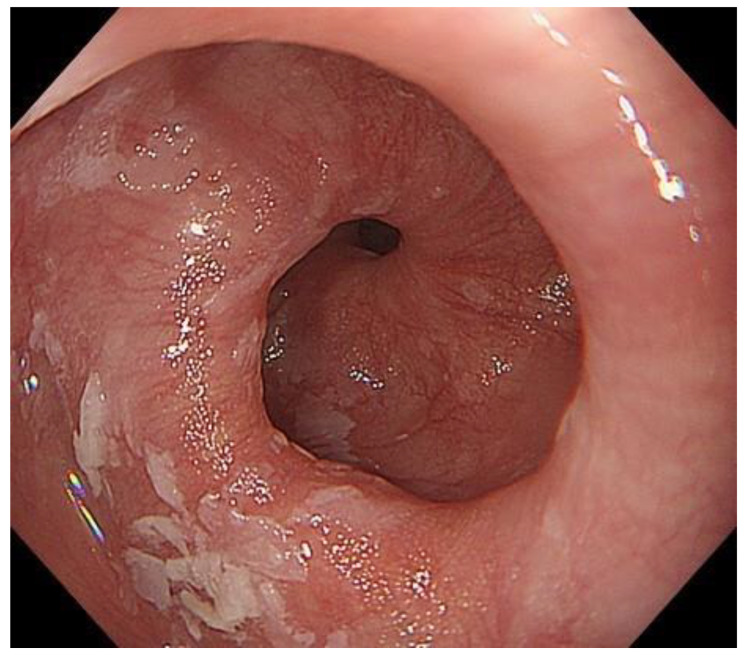
A case with hypercontractile esophagus in which corkscrew esophagus was observed.

**Figure 7 diagnostics-13-00695-f007:**
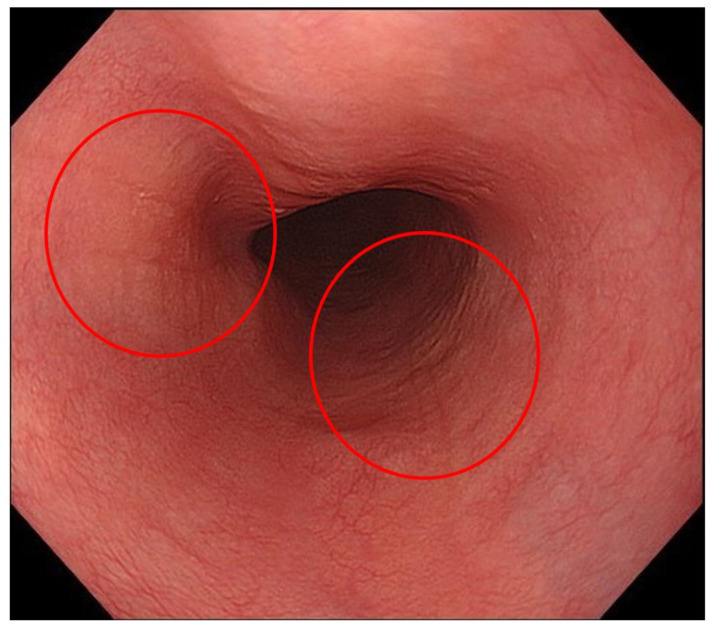
A case in which furrows were seen focally. Since furrows were seen focally (red circles), these findings might not be detected in white light imaging.

**Figure 8 diagnostics-13-00695-f008:**
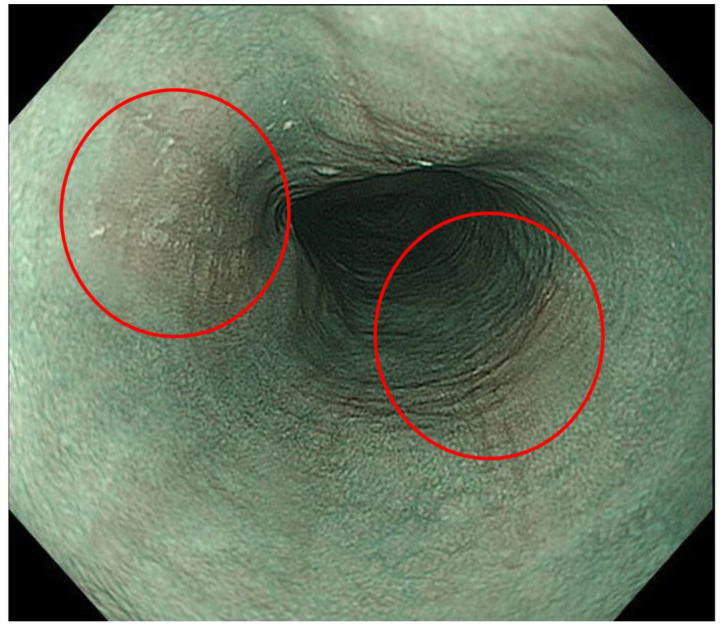
Beige mucosa was seen in narrow-band imaging. Beige mucosa could be observed in narrow-band imaging (red circles), and this finding made the detection of eosinophilic esophagitis easier than furrows in white light imaging in this case.

## Data Availability

No new data were created or analyzed in this study. Data sharing is not applicable to this article.

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
