# Peer review of "Usefulness of Endoscopy for the Detection and Diagnosis of Primary Esophageal Motility Disorders and Diseases Relating to Abnormal Esophageal Motility"

_diagnostics, 2023, doi:10.3390/diagnostics13040695_

Round 1
Reviewer 1 Report
The author reviewed well the endoscopic findings of primary esophageal motility disorders (EMDs) including achalasia and that of various diseases potentially inducing abnormal esophageal motility. Understanding endoscopic findings of functional disease, which are often underdiagnosed or misdiagnosed in clinical practice, is remarkably informative and educational for the reader. Few papers have summarized this content and is worth of publication.
EIPD, candida esophagitis, RE, BE, EoE can be generally diagnosed by endoscopy, and have been described by previous many reports including endoscopic findings. They are commonly classified into organic disease. Main contents of this review are “2.1. Endoscopic Findings Related to Primary EMDs.” In this section, the author introduced several characteristic endoscopic findings of achalasia and its related diseases such as Gingko leaf sign, Champagne glass pinstripe pattern, corona appearance corkscrew or rosary beads appearance, with no endoscopic image. The author should present as many of those endoscopic images as possible, rather than EoE images.
The author detailed described on not only the endoscopic findings of primary EMDs but those of organic diseases potentially inducing abnormal esophageal motility such as EIPD, candida esophagitis, RE, BE and EoE. Therefore, the title; Usefulness of Endoscopy for the Detection and Diagnosis of Esophageal Motility Disorders may not well fit to their main text.
Author Response
The author reviewed well the endoscopic findings of primary esophageal motility disorders (EMDs) including achalasia and that of various diseases potentially inducing abnormal esophageal motility. Understanding endoscopic findings of functional disease, which are often underdiagnosed or misdiagnosed in clinical practice, is remarkably informative and educational for the reader. Few papers have summarized this content and is worth of publication.
Comments:
- EIPD, candida esophagitis, RE, BE, EoE can be generally diagnosed by endoscopy, and have been described by previous many reports including endoscopic findings. They are commonly classified into organic disease. Main contents of this review are “2.1. Endoscopic Findings Related to Primary EMDs.” In this section, the author introduced several characteristic endoscopic findings of achalasia and its related diseases such as Gingko leaf sign, Champagne glass pinstripe pattern, corona appearance corkscrew or rosary beads appearance, with no endoscopic image. The author should present as many of those endoscopic images as possible, rather than EoE images.
We appreciate your valuable comments. Based on your comment we add endoscopic images related to primary EMDs as many as we can.
- The author detailed described on not only the endoscopic findings of primary EMDs but those of organic diseases potentially inducing abnormal esophageal motility such as EIPD, candida esophagitis, RE, BE and EoE. Therefore, the title; Usefulness of Endoscopy for the Detection and Diagnosis of Esophageal Motility Disorders may not well fit to their main text.
We appreciate your valuable comments. We changed the title as “Usefulness of Endoscopy for the Detection and Diagnosis of Primary Esophageal Motility Disorders and Diseases Relating to Abnormal Esophageal Motility”.

Reviewer 2 Report
Comment to authors
This manuscript reviewed the role of endoscopy in detection and diagnosis of esophageal motility disorders. The content of this manuscript was well described. However, there remain some concerns following the below comments.
Major comments:
1. In the introduction part, the explanation of POEM is too long.
This is a review article on the endoscopic detection and diagnosis of EMDs, and the description of treatment should be kept to a minimum, especially in the introduction part.
2. Esophageal motility disorders are classified as primary or secondary. Primary esophageal motility disorders are divided into esophageal achalasia and other spastic disorders. Before the section on detection and diagnosis, a section on the definitions of the EMDs discussed in this article should be included and mentioned.
3. As the author mentioned on Page 6, magnifying characteristics of EoE appear to be important. Several reports indicated characteristics other than ref 99. Please refer to those characteristics.
● Ichiya T, Tanaka K, Rubio CA, et al. Evaluation of narrow-band imaging sign in eosinophilic and lymphocytic esophagitis. Endoscopy 2017;49:429–37. (doi:10.1055/s-0043-101685.)
● Yasuda T, Yagi N, Omatsu T, et al. Magnifying Endoscopic Characteristics in Eosinophilic Esophagitis Patients. Techniques and Innovations in Gastrointestinal Endoscopy 2021; 107-109. (https://doi.org/10.1016/j.tige.2020.07.003)
Minor comments:
4. Please add some characteristic endoscopic images of the primary EMDs mentioned in the text.
5. The authors should describe the author's contributions on Page 8. Further, if the authors use images of EMDs from specific patients, they should describe the informed consent statement. (A general opt-out method would be fine.)
Author Response
This manuscript reviewed the role of endoscopy in detection and diagnosis of esophageal motility disorders. The content of this manuscript was well described. However, there remain some concerns following the below comments.
Major comments:
- In the introduction part, the explanation of POEM is too long. This is a review article on the endoscopic detection and diagnosis of EMDs, and the description of treatment should be kept to a minimum, especially in the introduction part.
We appreciate your valuable comments. We shortened explanations of POEM.
- Esophageal motility disorders are classified as primary or secondary. Primary esophageal motility disorders are divided into esophageal achalasia and other spastic disorders. Before the section on detection and diagnosis, a section on the definitions of the EMDs discussed in this article should be included and mentioned.
We appreciate your valuable comments. We noted a classification of EMDs in the introduction section.
- As the author mentioned on Page 6, magnifying characteristics of EoE appear to be important. Several reports indicated characteristics other than ref 99. Please refer to those characteristics.
- Ichiya T, Tanaka K, Rubio CA, et al. Evaluation of narrow-band imaging sign in eosinophilic and lymphocytic esophagitis. Endoscopy 2017;49:429–37. (doi:10.1055/s-0043-101685.)
- Yasuda T, Yagi N, Omatsu T, et al. Magnifying Endoscopic Characteristics in Eosinophilic Esophagitis Patients. Techniques and Innovations in Gastrointestinal Endoscopy 2021; 107-109. (https://doi.org/10.1016/j.tige.2020.07.003)
We appreciate your valuable comments. We referred these papers.
Minor comments:
- Please add some characteristic endoscopic images of the primary EMDs mentioned in the text.
We appreciate your valuable comments. Based on your comment we add endoscopic images related to primary EMDs as many images as we can.
- The authors should describe the author's contributions on Page 8. Further, if the authors use images of EMDs from specific patients, they should describe the informed consent statement. (A general opt-out method would be fine.)
We appreciate your valuable comments. We added information in Page 8.

Round 2
Reviewer 1 Report
This manuscript has been appropriately revised according to the reviewers' comments.